# Calcineurin-inhibitor free immunosuppression after lung transplantation – a single center case-control study in 51 patients converted to Mechanistic Target of Rapamycin (mTOR) inhibitors

Jens Gottlieb[1,2]\*, Bettina Fischer[1], Jonas C. Schupp[1], Heiko Golpon[1,2]

1 Respiratory Medicine, Hannover Medical School, Hannover, Germany, 2 German Center for Lung Research (DZL), Hannover, Germany

\* gottlieb.jens@mh-hannover.de

## Abstract

### Background

Data on calcineurin-inhibitor (CNI) free immunosuppression after lung transplantation (LTx) are limited. Aim of this study was to investigate CNI-free immunosuppression using mechanistic target of rapamycin (mTOR) inhibitors.

### Methods

This retrospective analysis was performed at a single center. Adult patients after LTx without CNI during the follow-up period were included. Outcome was compared to those LTx patients with malignancy who continued CNI.

### Results

Among 2,099 patients in follow-up, fifty-one (2.4%) were converted median 6.2 years after LTx to a CNI-free regimen combining mTOR inhibitors with prednisolone and an antimetabolite, two patients were switched to mTOR inhibitors with prednisolone only. In 25 patients, malignancies without curative treatment options were the reason of the conversion, with a 1-year survival of 36%. The remaining patients had a 1-year survival of 100%. Most common non-malignant indication was neurological complications (n = 9). Fifteen patients were re-converted to a CNI-based regimen. The median duration of CNI-free immunosuppression was 338 days. No acute rejections were detected in 7 patients with follow-up biopsies. In multivariate analysis, CNI-free immunosuppression were not associated with improved survival after malignancy. The majority of patients with neurological diseases improved 12 months after conversion. Glomerular filtration rate increased by median 5 (25 and 75% percentiles -6; +18) ml/min/1.73 m$^2$.

**Data Availability Statement:** All relevant data are within the paper and its Supporting Information files.

**Funding:** The author(s) received no specific funding for this work.

**Competing interests:** Jens Gottlieb: related to this work: Novartis (speaker fees) Jens Gottlieb unrelated to this work: Zambon /Breath Therapeutics (Research grant), Theravance (Advisory), Atheneum (Consultancy), Pierre Fabre (Advisory), Astra Zeneca (Speaker fee), Springer Healthcare (Advisory), Merck (Advisory), European Research Network (Consultancy), Precision (Advisory), German Center of Lung Research (Research grant), Deutsche Forschungsgemeinschaft (Research grant). Heiko Golpon, Bettina Fischer and Jonas Schupp have declared that no competing interests exist.

**Abbreviations:** CLAD, chronic lung allograft dysfunction; CNI, calcineurin inhibitors; GFR, glomerular filtration rate; LTx, lung transplantation; mTOR, mechanistic target of rapamycin; SARS-CoV-2, severe acute respiratory syndrome coronavirus type 2; WHO, world health organisation.

## Conclusions

mTOR inhibitor based CNI-free immunosuppression may be safely performed in selected patients after LTx. This approach was not associated with improved survival in patients with malignancy. Significant functional improvements were observed in patients with neurological diseases.

## Introduction

Calcineurin-inhibitors (CNI) are the mainstay of maintenance immunosuppression (IS) after solid organ transplantation. Unfortunately, the use of CNI agents is associated with adverse effects including metabolic disturbances (for example new onset of diabetes mellitus), risk of infection and kidney failure. CNI is associated with an increased risk of malignancy [1]. These associated comorbidities may negatively influence outcomes after transplantation. Alternative immunosuppressive protocols providing efficient suppression of the immune system without such side effects are therefore sought. In liver, heart and kidney transplantation, CNI-free regimens were studied in randomized controlled trials [2–5] and such strategies were associated with improved renal function and slowing progression in chronic allograft dysfunction. CNI-free immunosuppression is used in selected patients in non-pulmonary transplantation. In contrast to these experiences, data on CNI-free immunosuppression after lung transplantation are limited [6–15].

Various agents are used as an alternative in CNI-free settings after LTx including belatacept [8, 10, 12], basiliximab [7, 14, 16], photopheresis [13] or mechanistic target of rapamycin (mTOR) inhibitors [17]. During the last two decades, immunosuppressive drugs inhibiting the mechanistic target of Rapamycin pathway have received considerable attention in solid organ transplantation. Everolimus and sirolimus are the two mTOR inhibitors commonly used in transplantation. The mechanistic target of rapamycin signaling pathway is essential in the cell cycle of T-lymphocytes and vascular smooth muscle cells. Its blockade leads to inhibition of progress from G1 to S phase of the cell cycle and causes combined immunosuppressive and antiproliferative effects. New mTOR inhibitors have demonstrated anti-tumor activity in clinical studies, and mTOR inhibition may be capable of inhibiting malignant cells and essential tumor-development in transplant recipients [18]. After lung transplantation, 10% of all recipients are affected by malignancy. Non-melanotic skin cancers and lymphomas are the leading forms of cancer with an increased risk for most forms of epithelial cancers.

Aim of this study was to investigate indications, safety and outcome of CNI-free immunosuppression based on mTOR inhibitors in a large LTx cohort.

## Material and methods

A retrospective analysis at a single, specialized outpatient follow-up clinic for adult patients after lung transplantation was performed.

All adult patients with at least one visit in the LTx outpatient follow-up clinic since 1993 were screened and patients without a calcineurin-inhibitor (CNI) regimen on at least a single visit were identified and included. The study was conducted in accordance with the ethical guidelines of the 1975 declaration of Helsinki. Patients provided written informed consent for anonymized data analysis in retrospective studies within the German center of lung research.

The use of data to conduct retrospective analysis was covered by ethics committee´s vote (No 2923–2015, last update September 24th 2021).

The follow-up period ended on September 30th 2021 for all patients or at time of death, whichever occurred first.

In our program, standard maintenance immunosuppression consisted of a triple drug regimen including CNI, prednisolone and antimetabolites (mycophenolate mofetil, azathioprine). Occasionally, patients were switched from an antimetabolite to a proliferation signal inhibitor in case of e.g. leukopenia, kidney impairment or recurrent CMV infections. Target trough levels for patients more than 2 years after LTx were 6–10 ng/ml for tacrolimus and 50–90 ng/ml for cyclosporine.

For LTx patients with cancer after transplant, a change in immunosuppression was evaluated case by case depending on the risk of rejection, tumor recurrence, and patient preference. In most patients, antimetabolites for 6 months were paused while continuing CNI. In patients converted to a CNI-free protocol, an mTOR inhibitor (sirolimus or everolimus) was used instead of CNIs. Target trough levels for sirolimus and everolimus were 8–14 ng/ml and for mycophenolate > 1.5 ng/ml. Azathioprine was initially dosed at 1–2 mg/kg body weight aiming for a total lymphocyte count of 800–1,200 /ml. No other regimes of CNI-free immunosuppression were used during the study period. Reasons for conversion to CNI-free immunosuppression, complications and outcome after conversion were recorded.

Patients were followed in intervals ranging from two weeks to annual visits depending on the time after transplantation and clinical stability. Patients with new medical problems were usually seen in monthly or 3-monthly intervals. All patients were monitored by central drug monitoring in our blood chemistry laboratory—even between visits—by sending in blood samples drawn by local physicians. Case management by telephone or video consultations was offered to all patients between visits. At each appointment in the outpatient clinic history, physical exam, spirometry and laboratory tests including immunosuppressant levels were obtained. Bronchoscopy with bronchoalveolar lavage and transbronchial biopsy was performed routinely during the first year and whenever clinically indicated thereafter. Surveillance transbronchial biopsy after conversion to a new maintenance regimen and screening for donor-specific antibodies was performed since 2018. Routine testing for donor-specific antibodies was not performed after conversion. Presence of donor-specific antibodies was not a contraindication for conversion. Surveillance biopsies during the first postoperative year were performed since 2011 in our program excluding patients with unilateral LTx. Spirometry was performed according to American Thoracic Society/European Respiratory Society guidelines [19]. Any change in FEV1 of more than 10% from the previously recorded value led to prompt investigation for work up including bronchoscopy. The same approach was applied in case onset or worsening of hypoxemia or new opacities on imaging. Chronic lung allograft dysfunction (CLAD) was defined as persistent forced expiratory volume in 1 second (FEV1) <80% in relation to the baseline FEV1 according to recently established criteria [20]. Acute cellular rejection was defined according to published histopathological criteria on transbronchial biopsy [21].

Glomerular filtration rate was calculated according to the CKD-EPI formula [22] and values were compared at conversion to those 12 months after conversion. For patients with missing GFR at 12 months, the worst of either the last observed GFR prior to the missing visit or the next observed GFR was imputed, if available.

Cases of malignancies were classified into noncurative or curative treatment goals according to malignancy-specific guidelines (e.g. R0 resection without metastases (e.g. T1 lung cancer), curative radiotherapy, use of adjuvant or neoadjuvant protocols). The world health organization performance status scale ranging from 0 to 4, with 0 denoting perfect

health and 4 completely disability (bedbound) was used to assess functional performance in patients with neurological diseases. Re-conversion to CNI containing immunosuppression was performed after a tumor-free interval of at least 6 months, in case of side effects and according to patient preference.

For comparison, a cohort all LTx patients with malignancy (excluding superficial non-melanotic skin cancer) that continued CNI was identified in our program.

## Statistics

Statistical analysis was performed with metric variables expressed as medians and 25 and 75% quartiles and categorical variables by absolute numbers and percentage of data entries. Univariate analyses were performed using the Mann–Whitney test for continuous variables and chi-square test for categorical variables. Survival analysis was performed using the Kaplan–Meier method. For the multivariate survival analysis, LTx patients with malignancy were identified in our database. Patients with CNI-free immunosuppression were compared to those with continued CNI immunosuppression. Cox regression analysis was conducted to analyze survival after tumor diagnosis. Variable of interest of group comparison were included in the model were chosen according to clinically reasoning and included age, curative or palliative approach, CNI-free immunosuppression, tumor type and preexisting CLAD. The level of significance was set at $\leq 0.10$ for including variables identified by univariate analysis between groups.

## Results

Among 2,099 patients with any visit in our outpatient clinic, 51 (2.4%) were on a CNI-free immunosuppression regimen during follow-up (Fig 1). The first patient identified was converted in August 2006 and half of the patients were switched after October 2013.

In all 51 patients, CNIs were converted to a regimen containing an mTOR-inhibitor (n = 7 (14%) sirolimus, n = 44 (86%) everolimus). Forty-seven patients (92%) were converted to a triple drug protocol including prednisolone and an antimetabolite. In two patients each, mTOR inhibitors were used in a dual drug regimen with either an antimetabolite or a steroid. Patient demographics are displayed in Table 1. The first visit with a CNI-free regimen occurred median 6.1 years (25 and 75% percentiles 2.6; 6.6 years) after transplantation. Three patients had a pre-transplant cancer history (astrocytoma, hodgkin´s disease and cervix cancer), two of those became CNI-free with a new malignancy after LTx.

In 37 patients, malignancy was the reason for conversion. Most prevalent malignancy in CNI-free patients was lung cancer (n = 10, 27% incl. 1 case of small cell lung cancer), renal cancer (n = 4, 11%), invasive skin cancer (n = 5, 14% incl. 1 case of melanoma), colorectal cancer (n = 4, 11%, incl. 1 case with additional lung cancer), bladder cancer (n = 3, 8%), esophageal cancer (n = 3, 8%), lymphoma (n = 2, 5%) and miscellaneous (n = 7, 19%). The overall proportion of patients with lung cancer, gastrointestinal cancer or lymphoma (n = 19, 51%) was similar to our entire cohort of cancer patients (Fig 2 and S1 Table). In 12 CNI-free patients, anti-tumor therapy had a curative approach. Chemotherapy was used in the majority of cancer patients as treatment modality (n = 23), radiation was used in 12 and 16 patients underwent surgery for surgery.

Most common non-malignant indications for the CNI-free regimen were neurological complications with severe disability: severe disabling polyneuropathy (n = 4), encephalitis (n = 3 incl. two cases of JC-Virus infection), and one case each of Guillain-Barre-syndrome and thrombotic thrombocytopenic purpura with coma and seizures.

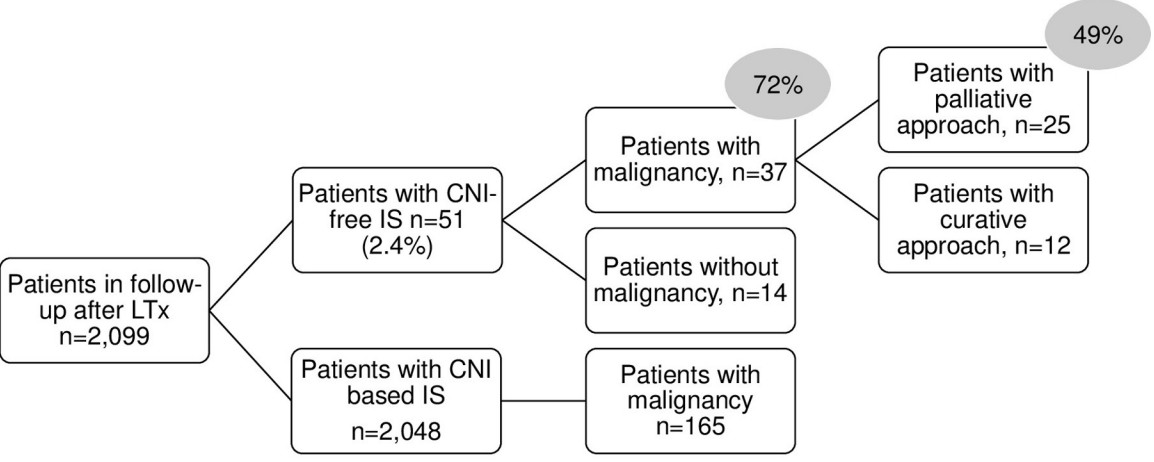

CNI – calcineurin inhibitor, LTx – lung transplantation, IS- immunosuppression

**Fig 1. Flowchart of patients.** CNI–calcineurin inhibitor, LTx–lung transplantation, IS–immunosuppression.

Median duration of CNI-free immunosuppression was 327 days (25 and 75% percentiles 120; 746 days) in patients with malignancy and median 755 days (25 and 75% percentiles 187; 2,192 days) in patients without malignancy as a reason for switching (entire cohort 338 days, 25 and 75% percentiles 120; 782 days). Fifteen patients (29%, n = 11 with malignancy) were re-converted to a CNI-containing regimen after median 199 days (25, 75% quartiles 97; 595 days). In additional five patients, temporal interruption of mTOR-therapy and intermittend reconversion to a CNI was observed.

## Outcome

Forty-one patients died during a follow-up of median 633 days after switch. The majority of deaths were caused by malignancy (Table 2). All tumor-associated deaths occurred in patients with a palliative approach and were caused by the same tumor. A single patient developed a new malignancy (acute leukemia) 446 days after conversion to CNI-free immunosuppression motivated by non-malignant disease and died from severe acute respiratory syndrome corona-virus type 2 infection. Fig 3 demonstrates outcome of the subgroup of CNI-free patients with malignancy (n = 37).

One-year survival in patients with CNI-free immunosuppression was 36, 100 and 100% in those with malignancy in a non-curative approach, with a curative approach and those with non-malignant indications for switch, respectively. One-year survival in those cancer controls (n = 165) with continued CNI-based immunosuppression was 39 and 82% in palliative and curative patients, respectively. In multivariate analysis, survival after malignancy was independently associated with younger age at diagnosis, a curative approach, and absence of CLAD (Table 3), while CNI-free immunosuppression had no effect.

For Thirty-seven patients, paired GFR calculated at conversion and 12 months after conversion was available. Median GFR increased by 5 (25 and 75% percentiles -6; +18), and 8 (25 and 75% percentiles -6; +21) ml/min/1.73m$^2$ if GFR was above 30 ml/min/1.73m$^2$ before conversion. Seven out of nine patients with neurological entities as motivation for CNI-free immunosuppression could be reviewed clinically after conversion. Functional improvement after 12

**Table 1. CNI-free patient characteristics of (n = 51).**

| | Calcineurin inhibitor-free immunosuppression n = 51 |
|---|---|
| Sex, n (%) | |
| Female | 22 (43) |
| Male | 29 (57) |
| Age at transplant, median years (25th, 75th percentile) | 48 (40, 56) |
| Switch after lung transplantation, median years (25th, 75th percentile) | 6.1 (2.6, 6.6) |
| Transplant procedure, n (%) | |
| Bilateral lung | 42 (82) |
| Unilateral lung | 5 (10) |
| Heart-lung | 3 (6) |
| Lung-liver | 1 (2) |
| Diagnosis, n (%) | |
| emphysema / alpha-1 antitrypsin deficiency | 11 (22) |
| Fibrosis / interstitial lung disease | 17 (33) |
| Cystic fibrosis / bronchiectasis | 13 (25) |
| Pulmonary hypertension / vascular diseases | 6 (12) |
| Other | 4 (8) |
| Pre-transplant history of malignancy Reason for switch, n (%) | 3 (6) |
| Malignancy | 37 (72) |
| Neurological diseases | 9 (18) |
| Sclerosing cholangitis | 1 (2) |
| Recurrent bronchostenosis | 4 (8) |
| Immunosuppression before switch, n (%) | |
| Calcineurin inhibitor + Steroid + antimetabolites | 42 (84) |
| Calcineurin inhibitor + Steroid | 2 (4) |
| Calcineurin inhibitor + Steroid + mechanistic target of rapamycin inhibitor | 3 (6) |
| Calcineurin inhibitor quadruple | 4 (8) |

months was observed in six patients by improvement in WHO performance status: one patient improved by 1 stage, one by 2 stages, three by 3 stages and one patient by 4 stages, respectively.

## Safety

Reasons for re-conversion in 15 patients were radiation therapy (n = 2), impaired wound healing (n = 1), diarrhea (n = 4), mucositis (n = 2), termination of adjuvant antitumor therapy (n = 2), seizure (n = 1), pneumonia / pneumonitis (n = 1), and pulmonary embolism (n = 1). Seven transbronchial biopsies were performed in six patients while on CNI-free immunosuppression. None of these biopsies revealed acute rejection grade A1 or higher. Forty-four (86%) of all recipients were hospitalized at some time point after conversion to CNI-free immunosuppression and 21 patients (41%) were hospitalized with an infection while on CNI-free immunosuppression. With the exception of 4 viral infections (influenza, metapneumovirus and severe acute respiratory syndrome coronavirus type 2) all infections were presumed of bacterial origin. Two patients died from infection on day 19 (pneumonia after chemotherapy) and day 131 (sepsis after chemotherapy) after conversion A single patient stopped mTOR-inhibitor due to recurrent infections.

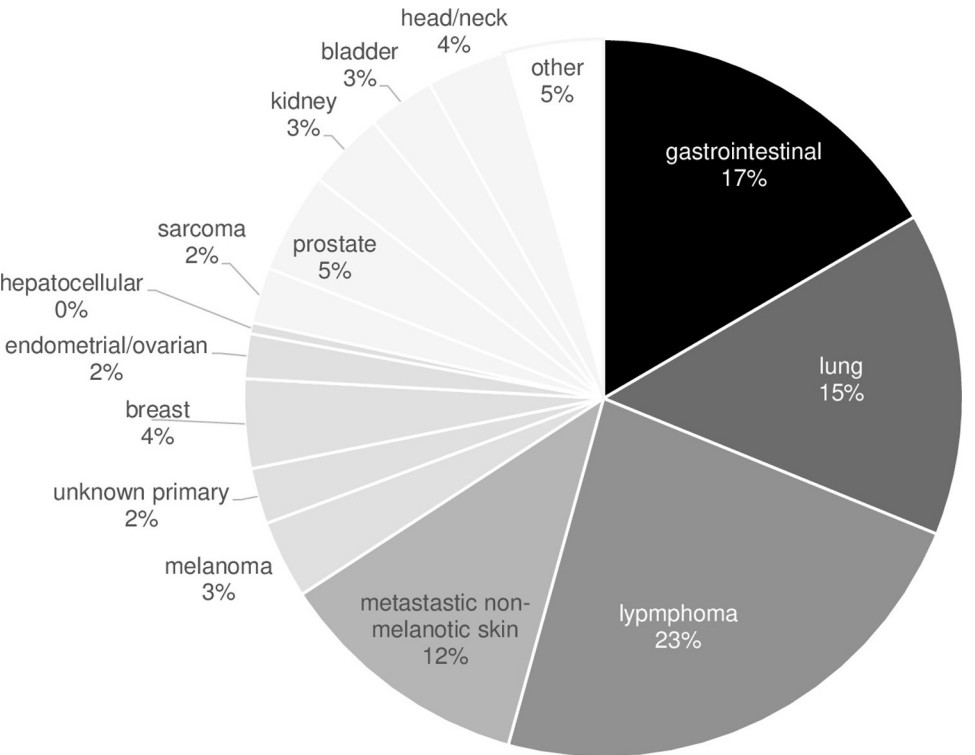

**Fig 2. Type of cancer in lung transplant recipients (n = 202).**

**Table 2. CNI-free patient outcome (n = 51).**

| | Calcineurin inhibitor-free immunosuppression n = 51 |
|---|---|
| Follow-up after switch, median days (25th, 75th percentile) | 633 (234, 2007) |
| Time on calcineurin inhibitor free immunosuppression, median days (25th, 75th percentile) | 338 (120, 782) |
| Immunosuppression after conversion, n (%) | |
| Sirolimus | 7 (14) |
| Everolimus | 44 (86) |
| Steroid | 49 (96) |
| Antimetabolite | |
| Mycophenolate Mofetil | 47 (92) |
| Azathioprine | 2 (4) |
| No antimetabolite (= proliferation signal inhibitor + Steroid) | 2 (4) |
| Re-conversion, n (%) | 15 (29) |
| Re-do-transplantation, n (%) | - |
| Deceased after conversion, n (%) | 41 (82) |
| Cause of death, n (%) | |
| Malignancy | 23 (56) |
| Unknown | 7 (17) |
| Chronic lung allograft dysfunction | 6 (15) |
| Infection | 2 (5) |
| Cardiovascular cause | 2 (5) |
| Trauma | 1 (2) |

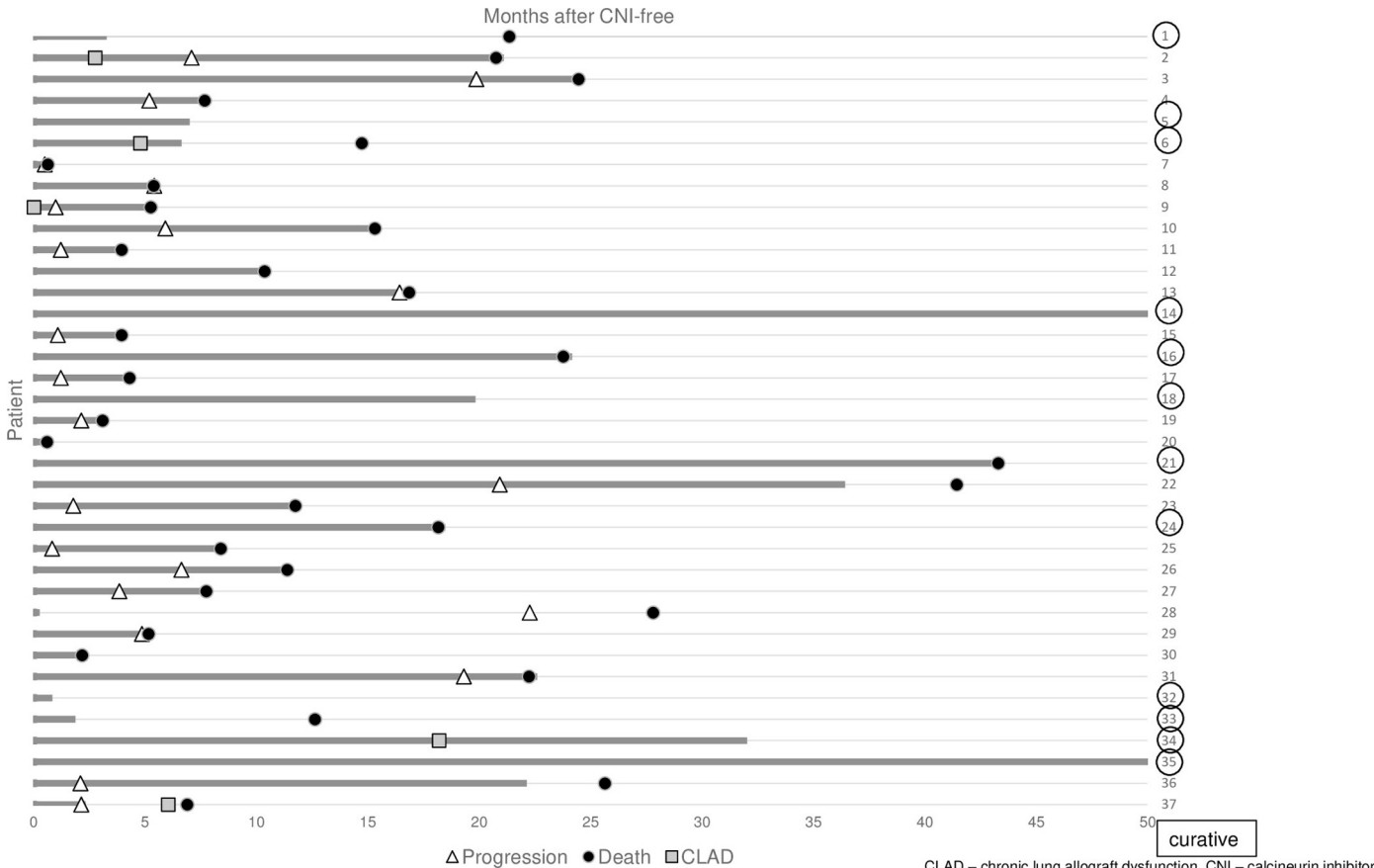

**Fig 3. Swimmer plot of outcome of CNI-free patients with malignancy (n = 37).** Grey bars represent time on CNI-free immunosuppression (patients #14 and #35 ongoing). CLAD–chronic lung allograft dysfunction, CNI–calcineurin inhibitor.

Seventeen patients had pre-existing chronic lung allograft dysfunction (CLAD) before conversion. Twenty-four patients (47%) had surveillance biopsies (60 biopsies in total) at any time before conversion. In four patients (in total eight biopsies), acute rejection with grade A1 or higher was revealed at some time before conversion. Ten patients were tested for donor-specific antibodies some time before conversion with four patients being positive on at least a single occasion.

Five patients developed new onset of chronic lung allograft dysfunction after conversion including three patients while on CNI-free immunosuppression 20, 515 and 555 days after conversion.

## Discussion

This study demonstrates that in selected LTx-patients, prolonged CNI-free immunosuppression based on an mTOR inhibitor regimen can be performed. Clinical improvement was observed in patients with neurological diseases. However, CNI-free immunosuppression was not associated with improved survival in de-novo malignancy, although statistical power was limited.

Everolimus, which was used most commonly in our series, is approved in some countries for treating certain breast cancers, neuroendocrine tumours, and renal cell carcinoma. It seems that the anti-cancer effect of mTOR inibitors is dose-dependent. However, the

**Table 3. Survival analysis in patients with malignancy (n = 202).**

| Covariate | N | Survival (months), median (95%-CI) | Univariate | | Multivariate | |
|---|---|---|---|---|---|---|
| | | | Hazard Ratio (95%-CI) | p | Hazard Ratio (95%-CI) | p |
| Age at malignancy diagnosis (years), median (25th, 75th percentile) | 202 | 20 (13, 27) | 1.026 (1.010, 1.042) | 0.002 | **1.020 (1.002, 1.038)** | **0.027** |
| Approach, n (%) | | | | | | |
| palliative approach | 107 | 8 (5, 11) | (Ref) | (Ref) | (Ref) | (Ref) |
| curative approach | 95 | 63 (37, 89) | 3.486 (2.422, 5.016) | <0.001 | **0.297 (0.200–0.440)** | **<0.001** |
| CNI-free, n (%) | | | | | | |
| no | 165 | 19 (9, 29) | (Ref) | (Ref) | (Ref) | (Ref) |
| yes | 37 | 20 (10, 30) | 1.179 (0.789, 1.762) | 0.422 | 0.898 (0.0.597-, 1.353) | 0.608 |
| Tumor type, n (%) | | | | | | |
| lymphoma | 48 | 30 (0, 63) | 1.057 (0.679, 1.647) | 0.806 | 1.381 (0.858, 2.223) | 0.183 |
| gastrointestinal | 33 | 13 (6, 24) | 1.569 (0.971, 2.536) | 0.066 | 1.039 (0.630, 1.714) | 0.881 |
| lung | 29 | 9 (2, 16) | 2.519 (1.569, 4.043) | <0.001 | 1.297 (0.784, 2.144) | 0.311 |
| other | 92 | 31 (16, 46) | (Ref) | (Ref) | (Ref) | (Ref) |
| CLAD (date onset before malignancy), n (%) | | | | | | |
| no | 134 | 28 (11, 45) | (Ref) | (Ref) | (Ref) | (Ref) |
| yes | 68 | 11 (7, 15) | 1.718 (1.225, 2.408) | 0.002 | **1.689 (1.184, 2.410)** | **0.004** |

CLAD–chronic lung allograft dysfunction, CNI–calcineurin inhibitors

everolimus doses used in tumor treatment is higher than the dose used to prevent transplant rejection as in our study and higher doses may be poorly tolerated in transplant patients.

There are limited cases published about switching from CNI-based immunosuppression to mTOR inhibitors in transplanted patients with malignancy. Alamo reported on 32 patients with malignancy (50% hepatocellular carcinoma) after liver transplantation switched to an mTOR inhibitor as CNI-free therapy [23]. During a follow-up of 16 months, survival was 69% under this regimen with a 15% tumor recurrence compared to our 1 year survival rate of 54% for all patients with malignancy. No graft rejections were observed in this Spanish study, and outcome was favourable in comparison to a control group of 130 patients continuing ciclosporine. A 52% discontinuation rate (19% due to infections) of mTOR inhibitors in combination or without calcineurin inhibitors was published in 147 patients in a single center retrospective analysis without reporting details in the CNI-free subgroup [15]. CNI-free immunosuppression after LTx based on mTOR inhibitors were presented as a conference abstract and included 12 long term patients (three with malignancy) converted to a CNI-free everolimus-based immunosuppressive regimen mainly to preserve kidney function [17]. Three patients had an acute rejection within a median follow-up of 1.6 years with no further details reported. A meta-analysis of randomized controlled trials of kidney transplant patients converted to sirolimus median 1.1 year after transplant demonstrated a reduction of cancer from 4.4% to 3.6% in comparison to a ciclosporine-based immunosuppression although the 4 year survival was lower for sirolimus (92% vs 95%) [24]. In kidney transplantation, there is some evidence that conversion from CNI to mTOR inhibitor could be associated with an increased risk of chronic rejection [25]. The same authors stated that the cancer preventing effects of mTOR inhibitors mainly concerns non-melanoma skin cancers and Kaposi sarcomas, both represented just 10% of our underlying malignant diseases. There is no clear evidence if and how immunosuppression in patients after solid organ transplantation with de-

novo malignancy should be adapted. The optimal approach to reducing immunosuppression in this setting is uncertain, and strategies vary depending upon cancer type and type of transplant. Individual decision making is recommended incorporating patient´s perspective [25].

Belatacept was used as an alternative agent to calcineurin inhibitors after lung transplantation. An open label randomized controlled trial was prematurely stopped after 5 out of 13 patients treated with belatacept in conjunction with mycophenolate and prednisolone died from various causes (CLAD, lymphoma, hematothorax, embolism) [8]. In 2 case series and 3 case reports in which belatacept was used in a CNI free protocol to improve kidney function, 22 patients in total were followed between 54 and 180 days [6, 9–12]. One year survival was 50%, and two patients died with acute rejection grade A4 and respiratory failure.

Basiliximab was reported in 2 cases as another option for CNI-free immunosuppression while 14 patients were continued on reduced CNI doses [7]. In 3 cases with posterior reversible encephalopathy, basiliximab was used for a couple of days to bridge CNI-holidays [14]. In a conference abstract Eiting et al reported 26 LTx patients with renal, hematologic and neurological CNI- toxicities using belatacept during CNI holidays. Toxicities resolved in 50% but no further details were reported [16].

In a single case report, photopheresis and immunoglobulins were used as stand-alone dual immunosuppression in a cystic fibrosis patient with M. abcessus infection for more than 27 months [13]. Case reports have described the absence of any immunosuppression in lung transplant recipients after bone marrow transplantation from the same donor [26, 27].

In general, tolerance of mTOR therapy was acceptable. Excluding patients with planned reconversion after adjuvant tumor therapy, 25% were reconverted to a CNI based protocol due to side effects. There was no specific side effect that was often reported as a reason for reconversion. The drop-out rate is in line with 10 to 26% due to side effects in randomized controlled trials [28, 29].

In our cohort, minor improvement of kidney function was noted after conversion to a CNI-free protocol. Irreversible kidney CNI-toxicity might explain this observation. In two studies with a median follow up of 5.6 and 6.6 years, quadruple immunosuppression after LTx with low-dose CNI in combination with mTOR inhibitors resulted in no GFR difference compared to conventionally dosed CNI-based immunosuppression [30, 31]. These findings suggest that chronic CNI renal toxicity is not dose dependent.

Limitations of our study are a single center retrospective analysis with a majority of long-term patients after LTx with malignancy. In our study, multiple types of cancer were included and effects of specific tumours could not be evaluated. In addition, mTOR inhibitors were started at varying intervals after LTx. Furthermore, the CNI free immunosuppression was initiated after the diagnosis of malignancy was made, thus not allowing for an analysis of the preventive" potential of mTOR inhibitors. Although donor-specific antibodies and acute rejections were present in a significant proportion of the included patients, there might be a selection bias of patients tolerant for a reduced immunosuppression. Many of the malignancies in study patients have not been associated with chronic use of immunosuppressive therapy or oncoviruses so the effect of mTOR inhibitors might be underestimated.

## Conclusions

In conclusion, mTOR inhibitor based CNI-free immunosuppression can be performed in selected patients after LTx, with every fourth patient being re-converted. In contrast to belatacept and basiliximab, long-term CNI-free therapy based on mTOR inhibitors has an acceptable safety profile with functional improvements of patients with neurological diseases. Our approach seems not be associated with improved survival in patients with malignancy. Minor

and inconsistent improvement in kidney function were observed after conversion. Earlier intervention might prove beneficial in the future and with lacking randomizied controlled trials centers should develop conversion protocols for CNI-free immunosuppression including regular sequential follow up assessments.

## Supporting information

**S1 Table. Characteristics in patients with malignancy (n = 202).**
(DOCX)

**S1 Dataset.**
(XLSX)

## Acknowledgments

Christina Valtin, Hannover.

## Author Contributions

**Conceptualization:** Jens Gottlieb, Jonas C. Schupp, Heiko Golpon.

**Data curation:** Jens Gottlieb, Bettina Fischer, Jonas C. Schupp, Heiko Golpon.

**Formal analysis:** Jens Gottlieb, Heiko Golpon.

**Investigation:** Jonas C. Schupp, Heiko Golpon.

**Methodology:** Jens Gottlieb, Jonas C. Schupp, Heiko Golpon.

**Project administration:** Jens Gottlieb, Heiko Golpon.

**Supervision:** Heiko Golpon.

**Validation:** Bettina Fischer.

**Visualization:** Jens Gottlieb.

**Writing – original draft:** Jens Gottlieb, Bettina Fischer, Jonas C. Schupp, Heiko Golpon.

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
