## [Decision Letter · Decision Letter 0]

31 Jan 2023

PONE-D-22-28204Calcineurin-Inhibitor Free Immunosuppression After Lung Transplantation – a Single Center Case-control Study in 51 Patients converted to Mammalian Target of Rapamycin (mTOR) InhibitorsPLOS ONE

Dear Dr. Gottlieb,

Thank you for submitting your manuscript to PLOS ONE. After careful consideration, we feel that it has merit but does not fully meet PLOS ONE’s publication criteria as it currently stands. Therefore, we invite you to submit a revised version of the manuscript that addresses the points raised during the review process.

We look forward to receiving your revised manuscript.

Kind regards,

H. Hakan Aydin, MD, FAACC

Academic Editor

PLOS ONE

Journal Requirements:

2. Please provide additional details regarding participant consent. In the ethics statement in the Methods and online submission information, please ensure that you have specified what type you obtained (for instance, written or verbal, and if verbal, how it was documented and witnessed). If your study included minors, state whether you obtained consent from parents or guardians. If the need for consent was waived by the ethics committee, please include this information

Reviewers' comments:

Reviewer's Responses to Questions

**Comments to the Author**

1. Is the manuscript technically sound, and do the data support the conclusions?

Reviewer #1: Partly

Reviewer #2: Yes

2. Has the statistical analysis been performed appropriately and rigorously? 

Reviewer #1: Yes

Reviewer #2: I Don't Know

3. Have the authors made all data underlying the findings in their manuscript fully available?

Reviewer #1: Yes

Reviewer #2: Yes

4. Is the manuscript presented in an intelligible fashion and written in standard English?

Reviewer #1: Yes

Reviewer #2: Yes

5. Review Comments to the Author

Reviewer #1: There are several problems with this manuscript:

- This is a retrospective analysis rather than prospective study and the mTOR inhibitor was started at varying intervals for the treatment group

- The group of patients with malignancies is very small (51, compared to 2049)

- Multiple types of cancer are being compared together

- The CNI free immunosuppression was initiated after the diagnosis of malignancy was made, thus not allowing for an analysis of the "preventive" potential of mTOR inhibitors used as CNI free immunosuppression

mTOR inhibitors have been used in Lung transplant programs for many years, although this reviewer acknowledges that they are seldom used in an elective or protocolized fashion. Effectiveness of immunosuppression is well established, so there is nothing novel in reporting that they are well tolerated. Unfortunately, this study does not help to answer the malignancy question as the immunosuppression was changed "after the fact". A better approach would be to develop a protocolized conversion timeline, with regular sequential follow up.

Reviewer #2: I was delighted to have a chance to peer review the submission to PLOS ONE entitled “Calcineurin-Inhibition Free Immunosuppression After Lung Transplantation – a Single Center Case-Control Study in 51 Patients converted to Mammalian Target of Rapamycin (mTOR)”, submitted by the Hannover group. In it, you report on a retrospective analysis of 51 patients over almost 30 years at a single center where patients had their CNI stopped and an alterative regimen was used. Your aim was to “investigate indications, safety, and outcomes of CNI-free immunosuppression based on mTOR inhibitors in the cohort” (last sentence of the introduction). About half of them were patients who had been changed because of active malignancies. Some later had CNI’s re-introduced.

You describe better than expected outcomes (in my estimation), with no acute rejections reported (but limited numbers having had biopsies), high mortality for the oncology patients (1-year survival of 36%), low mortality for the non-oncology patients (1-year survival 100%), improvement in neurologic side effects for those who had been changed for that reason, and a modest improvement in GFR for the cohort. The median duration of CNI-free immunosuppression was 338 days, with the period beginning a median of 6.1 years after transplantation.

1. Why were there not more biopsies done after the switch?

2. Other than “CLAD or not”, were there any assessments of graft health after the switch? What happened to the spirometry?

3.

You state in the results section that only 47% of the cohort of 51 patients had been biopsied at any time before the CNI free conversion. This is surprising to me and may need clarification. Are not all of your center’s patients getting transbronchial biopsies in their first year after transplant? If not, you may want to mention somewhere that this is your routine.

You state that 10 patients tested positive for donor specific antibodies before conversion. Please mention how you handled this information in general (did it influence decisions about converting to CNI-free?). Were patients routinely (or occasionally ad hoc) testing for DSAs after the conversion?

Your figures should each have a text box for the document with a description of what is in the figure.

It would be interesting and visually pleasing to see survival curves (ie Kaplan-Meier)

Under “2.2 Safety”, you describe incidence of hospitalization and infections in the group that was CNI free. How did this compare to patients who were on more standard therapy? You might consider a broader comparison looking at all patients (not just oncology patients) and outcomes with and without the conversion to CNI-free.

Minor:

1. In the first paragraph of the introduction, the word “haltering” was used. This is an irregular term and I might suggest using a different word such as “slowing”.

2. In the last sentence of the first paragraph of the introduction, change “CNI-free immunosuppression after lung transplantation is limited” to “CNI-free immunosuppression after lung transplantation are limited”.

3. Throughout the document, mTOR is referred to as “mammalian target of Rapamycin”. A more updated and common terminology is now “mechanistic target of Rapamycin”. Consider a change if you are inclined to agree.

4. The word “intermitted” just before section “2.1 outcomes” is an irregular word, please change.

5. In the “2.1 Outcomes” section, consider changing the description of “corona virus disease 2019” to “SARS CoV-2 infection”.

6. In Discussion, “Ina conference abstract Eiting” should be, “In a conference abstract Eiting”

7. In Discussion, “In our cohort, Minor improvement” should be, “In our cohort, minor improvement”

6. PLOS authors have the option to publish the peer review history of their article (what does this mean?). If published, this will include your full peer review and any attached files.

Reviewer #1: No

Reviewer #2: No

---

## [Author Response · Author response to Decision Letter 0]

8 Feb 2023

See detailed P2P response document

---

## [Decision Letter · Decision Letter 1]

15 Mar 2023

PONE-D-22-28204R1Calcineurin-Inhibitor Free Immunosuppression After Lung Transplantation – a Single Center Case-control Study in 51 Patients converted to Mechanistic Target of Rapamycin (mTOR) InhibitorsPLOS ONE

Dear Dr. Gottlieb,

Thank you for submitting your manuscript to PLOS ONE. After careful consideration, we feel that it has merit but does not fully meet PLOS ONE’s publication criteria as it currently stands. Therefore, we invite you to submit a revised version of the manuscript that addresses the points raised during the review process.

We look forward to receiving your revised manuscript.

Kind regards,

H. Hakan Aydin, MD, FAACC, EuSpLM

Academic Editor

PLOS ONE

Reviewers' comments:

Reviewer's Responses to Questions

**Comments to the Author**

1. If the authors have adequately addressed your comments raised in a previous round of review and you feel that this manuscript is now acceptable for publication, you may indicate that here to bypass the “Comments to the Author” section, enter your conflict of interest statement in the “Confidential to Editor” section, and submit your "Accept" recommendation.

Reviewer #2: All comments have been addressed

Reviewer #3: (No Response)

Reviewer #4: (No Response)

2. Is the manuscript technically sound, and do the data support the conclusions?

Reviewer #2: Yes

Reviewer #3: Partly

Reviewer #4: Partly

3. Has the statistical analysis been performed appropriately and rigorously? 

Reviewer #2: I Don't Know

Reviewer #3: Yes

Reviewer #4: No

4. Have the authors made all data underlying the findings in their manuscript fully available?

Reviewer #2: Yes

Reviewer #3: Yes

Reviewer #4: No

5. Is the manuscript presented in an intelligible fashion and written in standard English?

Reviewer #2: Yes

Reviewer #3: Yes

Reviewer #4: No

6. Review Comments to the Author

Reviewer #2: Thank you for your work in amending your submission. It is improved with your address of reviewers' comments.

Reviewer #3: I reveiwed a revised version of the manuscript entitled "Calcineurin-Inhibitor Free Immunosuppression After Lung Transplantation – a Single Center Case-control Study in 51 Patients converted to Mechanistic Target of Rapamycin (mTOR) Inhibitors" by Gttlieb et al.

First of all I have to state that I did NOT review the previous version.

Due to the nature of the study (retrospective single-center study) it is difficult to draw any conclusion.

For those patients who were converted from CNI-based IS to mTOR inhibitor-based IS the trough levels of sirolimus and everolimus are not mentioned.

On average after conversion from CNI- to mTOR-I-based IS how many times the patients were seen in the outpatient clinic?

mTOR-Is-associated side effects are poorly reported: when it occurred did it result in reverse conversion to CNI-based IS?

The paper is more suited for a pulmonology journal. Otherwise it could be considered as a Letter to the Editor.

Reviewer #4: Gottlieb and coll described their experience of CNI-free IS in lung transplant recipients.

In my opinion, the manuscript should be reoriented toward the main result of the study, ie the safety regarding regarding rejection, in lung transplant recipients with an advanced cancer.

In a methodological point of vue, I think the comparison with patients under CNI that presented different cancer types, and different cancer stages prevents analysis of patient survival grouped by IS

Other minor remarks:

* introduction: the statement that CNI free IS is a reality in the kidney and heart or pancreas transplant setting should be tempered. In fact all of these CNIfree protocols were stopped and antimetabolite are switched to everolimus in association with CNI.

A larger presentation of the burden of cancer after lung transplantation, and specificities of cancer in the lung transplantation could improve the intro

* Patients and Method

The authors stated that they used only nonparametric tests for quantitative comparisons, so with different different distribution in a small cohort. This suggest that cox-regression model to assess predictive factors is not really useful in this case-series. I suggest to remove this part, or at least to develop the statistical section to explain the number of variables included in the model, and the number of events per variables

* results: treatments used for cancer in association with mTOR should be developed to better understand the risk for rejection in this cohort

7. PLOS authors have the option to publish the peer review history of their article (what does this mean?). If published, this will include your full peer review and any attached files.

Reviewer #2: No

Reviewer #3: No

Reviewer #4: No

---

## [Author Response · Author response to Decision Letter 1]

23 Mar 2023

Please see separate P2P response document.

I added text of this table document to be sure:

PONE-D-22-28204R1: Title: “Calcineurin-Inhibitor Free Immunosuppression After Lung Transplantation – a Single Center Case-control Study in 51 Patients converted to Mechanistic Target of Rapamycin (mTOR) Inhibitors"

EMID:9ed5341f17931db9

*Page and line numbers refer to the marked version of the revised version of the manuscript

Editorial Office

# Requirements and prompts Response*

1 A rebuttal letter that responds to each point raised by the academic editor and reviewer(s). You should upload this letter as a separate file labeled 'Response to Reviewers'. A detailed Point-to-point response was added to the re-submission of the revised version of the manuscript.

2 A marked-up copy of your manuscript that highlights changes made to the original version. You should upload this as a separate file labeled 'Revised Manuscript with Track Changes'. A marked version of the revised version with tracked changes was uploaded. 

Label 'Revised Manuscript with Track Changes'.

3 An unmarked version of your revised paper without tracked changes. You should upload this as a separate file labeled 'Manuscript'. A unmarked version of the revised version without tracked changes was uploaded.

Label 'Manuscript'

Reviewer 2

# Reviewer Comment Response*

1 Thank you for your work in amending your submission. It is improved with your address of reviewers' comments.

 Thank you for the compliment.

Reviewer 3

# Reviewer Comment Response*

 1 I reveiwed a revised version of the manuscript entitled "Calcineurin-Inhibitor Free Immunosuppression After Lung Transplantation – a Single Center Case-control Study in 51 Patients converted to Mechanistic Target of Rapamycin (mTOR) Inhibitors" by Gttlieb et al.

First of all I have to state that I did NOT review the previous version.

Due to the nature of the study (retrospective single-center study) it is difficult to draw any conclusion. Thank you for clarifying this. For authors it is very difficult to deal with new reviewers without having knowledge of previous track changes. We´ll do our best to comply.

2 For those patients who were converted from CNI-based IS to mTOR inhibitor-based IS the trough levels of sirolimus and everolimus are not mentioned. Target trough levels are mentioned in the Materials and Methods section of the revised version of the manuscript on page 4 line 88. With more experience with mTOR based CNI-free immunosuppression we lowered our target trough levels to 8-12 ng/ml in recent years

3 On average after conversion from CNI- to mTOR-I-based IS how many times the patients were seen in the outpatient clinic Patient were postoperatively seen in 4-6 weeks interval in the first year, every three months in the second year, every six months up to year 5 and yearly thereafter Patients with new medical problems were usually seen in monthly or 3-monthly intervals. This information is included in the Materials and Methods section of the revised version of the manuscript (page 4, line 106-107).

4 mTOR-Is-associated side effects are poorly reported: when it occurred did it result in reverse conversion to CNI-based IS? Side effects of mTOR-IS are reported the results section of the revised version of the manuscript (page 13, line 226-230). 11 patients were reconverted to CNI free immunosuppression due to side effects; 4 patients due to other reasons. Most frequent side effects leading to termination of CNI-free IS were GI complaints (diarrhea, mucositis, together 66%). 21 patients were hospitalized due to infections on mTOR-IS and in a single patient mTOR-IS was terminated due to recurrent infection 

5 The paper is more suited for a pulmonology journal. Otherwise it could be considered as a Letter to the Editor.

 According to previous editorial and reviewer comments the paper tracked further as an original publication. 

Reviewer 4

# Reviewer Comment Response*

1 Gottlieb and coll described their experience of CNI-free IS in lung transplant recipients.

In my opinion, the manuscript should be reoriented toward the main result of the study, ie the safety regarding regarding rejection, in lung transplant recipients with an advanced cancer. We agree that safety information should be more strengthened and elaborated the results on biopsy results in lines 238-242 in the results section of the revised version of the manuscript on pages 13/14.

2 In a methodological point of vue, I think the comparison with patients under CNI that presented different cancer types, and different cancer stages prevents analysis of patient survival grouped by IS We agree with this comment and this information is contained under limitations page 17, line 296-297. 

3 minor introduction: the statement that CNI free IS is a reality in the kidney and heart or pancreas transplant setting should be tempered. In fact all of these CNIfree protocols were stopped and antimetabolite are switched to everolimus in association with CNI. The statement was tempered (introduction, page 3, line 63) 

4 minor A larger presentation of the burden of cancer after lung transplantation, and specificities of cancer in the lung transplantation could improve the intro. Informations about the burden of cancer was included on page 3. Lines 75-77.

5 minor Patients and Method

The authors stated that they used only nonparametric tests for quantitative comparisons, so with different different distribution in a small cohort. This suggest that cox-regression model to assess predictive factors is not really useful in this case-series. I suggest to remove this part, or at least to develop the statistical section to explain the number of variables included in the model, and the number of events per variables The requested information including the variables of interest of the cox regression model are listed in the stastitics section page 7, lines 145-147. 

6 minor results: treatments used for cancer in association with mTOR should be developed to better understand the risk for rejection in this cohort Chemotherapy was used in the majority of cancer patients (n=23), radiation was used in 12 and 16 patients underwent surgery for surgery. This information was added to the results section (page 9, lines 171-173)

---

## [Editor Report · Decision Letter 2]

5 Apr 2023

Calcineurin-Inhibitor Free Immunosuppression After Lung Transplantation – a Single Center Case-control Study in 51 Patients converted to Mechanistic Target of Rapamycin (mTOR) Inhibitors

PONE-D-22-28204R2

Dear Dr. Gottlieb,

We’re pleased to inform you that your manuscript has been judged scientifically suitable for publication and will be formally accepted for publication once it meets all outstanding technical requirements.

Kind regards,

H. Hakan Aydin, MD, FAACC

Academic Editor

PLOS ONE

---

## [Editor Report · Acceptance letter]

9 May 2023

PONE-D-22-28204R2 

Calcineurin-Inhibitor Free Immunosuppression After Lung Transplantation – a Single Center Case-control Study in 51 Patients converted to Mechanistic Target of Rapamycin (mTOR) Inhibitors 

Dear Dr. Gottlieb:

I'm pleased to inform you that your manuscript has been deemed suitable for publication in PLOS ONE. Congratulations! Your manuscript is now with our production department. 

Kind regards, 

on behalf of

Professor H. Hakan Aydin 

Academic Editor

PLOS ONE